# Calcium, Potassium, and Magnesium Affect the Nutritional Value of Tomato Grafted Fruits Grown in a Nutrient Film Technique System

**Rocío Maricela Peralta Manjarrez** [1], **Rafael Delgado Martínez** [2], **Adalberto Benavides Mendoza** [3], **Antonio Juárez Maldonado** [4] **and Marcelino Cabrera De la Fuente** [3,*]

1 CONAHCYT Posdoctoral, Departament of Horticulture, Universidad Autónoma Agraria Antonio Narro, Saltillo 25315, Coahuila, Mexico; rocio.peralta@uaaan.edu.mx

2 Faculty of Engineering and Sciences, Universidad Autónoma de Tamaulipas, Victoria City 87149, Tamaulipas, Mexico; rdelgado@docentes.uat.edu.mx

3 Departament of Horticulture, Universidad Autónoma Agraria Antonio Narro, Saltillo 25315, Coahuila, Mexico; adalberto.benavides@uaaan.edu.mx

4 Departament of Botany, Universidad Autónoma Agraria Antonio Narro, Saltillo 25315, Coahuila, Mexico; antonio.juarez@uaaan.edu.mx

* Correspondence: marcelino.cabrera@uaaan.edu.mx

**Abstract:** *Solanum lycopersicum* is a vegetable with a high mineral, nutraceutical, and vitamin content. It is a basic ingredient in the human diet, and its use is diverse within the kitchen. Grafting and Ca, K, and Mg applications can be used to increase production and raise the mineral contents of tomato fruit. An experiment was established with tomato plants using the "CID F1" variety with the "FORTAMINO" rootstock, established in the NFT system, determining the influence on the agronomic yield and mineral composition of the fruit. Grafted and non-grafted plants were considered, with five concentrations (0-0-0, 9-0-0, 0-12-0, 0-0-9, and 9-12-9 mEq L$^{-1}$) of Ca, K, and Mg, respectively. A highly significant difference was obtained in the grafted plants on high plants, number of leaves, number of fruits, polar diameters of fruits, equatorial diameters of fruits, and weight of fruits, with an increase in variables, FW 19% and NF 18%, and an improvement for the mineral composition in elements such as Ca 10%, P 1%, Mn 6%, Zn 7%, Cu 6%, Fe 64%, K 21%, and Mg 19%. The 9-12-9 meq formula improved Ca 6%, P 4%, Mn 12%, Zn 14%, Cu 8%, Fe 74%, and Mg 25%. The graft and the addition of calcium, potassium, and magnesium increased the mineral content in tomato fruits and improved the agronomic performance of the plants.

**Keywords:** graft; mineral composition; productivity; *Solanum lycopersicum*

## 1. Introduction

The tomato (*Solanum lycopersicum* L.) is an important source of bioactive compounds beneficial to human health, and 180 million tons are produced worldwide, of which 20% are used for industrial transformation [1]. This vegetable also participates in providing humans with a wide variety of nutrients with potentially beneficial effects. Its consumption favors the intake of nutraceuticals due to their nutritional and pharmaceutical properties, given their richness in lycopene and biologically active metabolites that are beneficial to human health [2]. This vegetable increases antioxidant status and is associated with high mineral uptake; low mineral intake can induce inflammatory processes and affect defense systems [3].

Grafting confers resistance to root problems in plants and tolerance to abiotic stresses, which is why the use of alternative techniques has been adopted in tomato production that will guarantee vegetative and root development for efficient nutrient absorption and make the productive system more efficient [4–7]. Grafted tomatoes have been shown to modify the general mineral composition of the plant and improve yield and biomass

accumulation [4,8,9]. Furthermore, nutrient uptake in horticultural crops is enhanced by the selection of suitable rootstocks, which play a vital role in manipulating the nutritional status of shoots by directly affecting ion uptake and transport [10].

Calcium, potassium, and magnesium are essential elements that, When added supplementally, intervene in the formation, production, and quality of tomatoes' fruits [9,11,12]. Ca modifies the total photochemical content and the concentration of lycopene in tomato fruits [11], as it is one of the essential elements for plant growth and fruit development [13]. The nutraceutical and mineral quality of the fruits is increased by the addition of potassium. It is a demanded element during the formation of tomato fruits [14]; the potassium-to-calcium ratio increases the quality attributes. Potassium participates in photosynthesis, respiration, and the activation of enzymes, and it has a significant influence on both the growth and the quality of fruits and vegetables [9,14]. Mg is involved in the photosynthetic reaction; it is a basic component of chlorophyll and activates enzymes that intervene in the synthesis of nucleic acids. The addition of this element contributes to the content of sugars and vitamin C, producing high-quality plants since it is an auxiliary in the metabolism of phosphates, plant respiration, and the activation of various enzyme systems involved in energy metabolism [12,15].

Therefore, grafting efficient nutrient absorption maximizes fertilization efficiency, increases plant vigor, ensures better yield and increases fruit quality [4,6]. The use of the grafting technique in tomato cultivation with the supplementation of Ca, K, and Mg is a technical–scientific novelty; it is considered a viable alternative to increase the quality of the crop and production. Its implementation, according to the formula recommended in the fruiting stage (9-12-9), could generate an improvement in the growth and development of plants.

Based on this, the objective of the present work was to evaluate the effect of grafting and the applications of Ca, K, and Mg to increase production and increase the mineral content of tomato fruit.

## 2. Materials and Methods

The research was carried out in a medium-technology micro-tunnel greenhouse at the horticulture department of the Antonio Narro Autonomous Agrarian University, Saltillo, Coahuila, Mexico. Latitude is $25°21'23.4''$ and longitude $101°02'10.6''$, with an altitude of 1760 m above sea level, where the average temperature is 25 °C.

### 2.1. Vegetal Material Used and Growing Conditions

Seeds of "Cid F1", from Harris Moran® produce an indeterminate tomato with uniform fruit size and shape, thick walls that provide excellent firmness, intense red fruits with long shelf life and high yield and these seeds work well in hydroponic systems. The rootstock Fortamino tomato was used, which is a tomato with a very high vigor and resistance to *Fusarium oxysporum* f.sp. *lycopersici*, races 0.1 and 2, and it is recommended for saline soils (Enza Zaden®, Enza Zaden®, Culiacan, Sinaloa, Mexico). Sowing was performed in a germinating tray of polystyrene (cavity measures: width 25 mm, length 25 mm, and depth 70 mm). Inside the greenhouse, there was an average irradiance of 187.47 W/m², the average temperatures recorded were 25 to 35 °C, and the average relative humidity inside ranged between 40 and 60%. The scion was sown on 19 February 2021 and the rootstock was sown 7 days later. This difference in sowing times was because the rootstock has greater vegetative vigor compared to the scion; with this, it was sought that both the stems of the scion and the rootstock had a similar diameter, which benefits the union of both plant structures. The grafted plants were placed in 250 mL Styrofoam containers. The substrate for filling the sowing trays was a mixture of peat moss with perlite at a 70:30 ratio.

*2.2. Graft*

The grafting process was carried out on 23 March 2021, when the plants had a 3 mm stem diameter, approximately, which was obtained 32 days after sowing of the scion. The splicing technique was used, which consisted of a cut at a 45° angle in the two plants to form one, and a silicone clip was applied to the union site. Subsequently, the grafted plants were kept for 15 days in an acclimatization chamber; they were in dark conditions for a period of 3 days, 5 with 50% shade and the rest of the days without shade. The relative humidity was 95%, and the temperature ranged between 25 and 35 °C for 10 days to favor the grafting process. Subsequently, the silicone clip was completely removed since the union between the scion and the rootstock had been produced and healed. The Steiner nutrient solution was used in mEq L$^{-1}$ (chemical composition: $NO_3^-$, $H_2PO_4^-$, $K^+$, $Ca^{++}$, $Mg^{++}$, $NH_4^+$, and $SO_4^=$) in different concentrations: approximately 25% at the beginning of vegetative growth (3, 0.6, 1, 3, 1, 0.5, 1.6 mEq L$^{-1}$, respectively); 50% vegetative growth at the beginning of flowering (8, 1, 4, 7, 3, 1, 4 mEq L$^{-1}$, respectively), 75% fruit set (12, 1.2, 6.5, 9, 3.5, 1, 6 mEq L$^{-1}$, respectively), and 100% ripening and harvest (13, 1.2, 7.5, 9, 4, 1, 7 mEq L$^{-1}$, respectively). With a pH of 6–6.5 and EC from 750 to 1500 mg L$^{-1}$, additional treatments were supplementary foliar applications of Ca, K, and Mg (9-12-9 mEq) to grafted and non-grafted plants.

*2.3. Nutrient Film Technique (NFT)*

The NFT system (Figure 1) used consisted of 6 PVC pipes 15.24 cm in diameter, with 16 holes 6 cm wide and 30 cm between holes. The distance between tubes was 60 cm, 68 L plastic boxes were used to collect the nutrient solution, and 25-watt submersible electric pumps for fish tanks with a circulation capacity of 1500 L per hour were used for recirculation of the nutrient solution. In a closed system, it is essential to maintain adequate EC during the growing season, renew the nutrient solution every time it reaches a value of 3.5 dS·m$^{-1}$, replace the transpired water with tap water, compensate for the transpired water with standard nutrient solution, and add all the nutrients according to the estimated consumption.

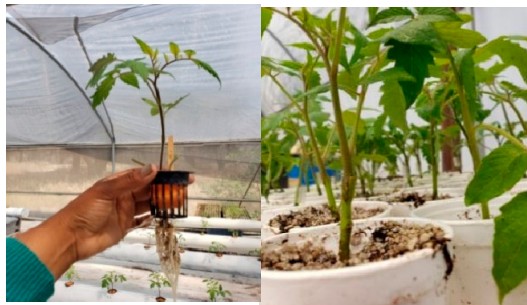

**Figure 1.** Plants established in greenhouse conditions.

*2.4. Transplant to the NFT System*

The transplant was carried out on 20 April (28 days after grafting), when the clips for the attachment had already been removed, with bare roots by completely removing the substrate from the root and immersing it in a 3% hydrogen peroxide solution as a preventive treatment against pathogenic microorganisms. Hydroponic baskets of three inches were used to adjust the plant inside the basket. A sponge was used to cover the root from direct sunlight, leaving the root free to have contact with the nutrient solution.

*2.5. Treatments Application*

The fertilizers that were added were the commercial products AMIFOL K®, HU-MISOIL MG 14®, and HUMISOIL CA 16® (Tradecorp, Zapopan, Jalisco, Mexico). Supplementary applications were applied by spraying direct spraying on the fruit. Two factors were considered: factor 1 consisted of grafting with two levels (with and without grafting);

factor 2: five levels corresponding to plants treated with different doses of Ca, K, and Mg (0-0-0, 9-0-0, 0-12-0, 0-0-9, 9-12-9), with a total of 10 treatments with 10 replicates. The treatments were evaluated using a $2 \times 5$ factorial arrangement in a completely randomized design with 10 experimental units per treatment, with a total of 100 plants.

### 2.6. Analysis of Crop Parameters

The productive cycle of the plants lasted until the development of the fifth floral cluster, when the plants emitted said cluster, and the apex was cut to stop it. At 110 days after transplanting, plant height (PH) was measured. This parameter was evaluated with a TOOLCRAFT® model TC0234 (Toolcraft, Nuevo Leon, Mexico) measuring tape with a three-meter capacity, measuring from the base of the stem to the apex of the stem. From the transplant, the number of leaves (NL) was counted every 15 days until the end of the growth cycle, and 10 repetitions per treatment were evaluated (n = 10). The fruits were harvested 82 days after transplanting, and 5 harvests were carried out in total, based on the USDA classification, which indicates that the fruits should have 90% of the surface of a uniform red color. The number of fruits (NF) was counted and weighed on a portable scale (OHAUS®, model H-8109, OHAUS Latinoamierica, Mexico DF, Mexico) to estimate the total yield. A total of 6 fruits were selected per treatment at the time of harvest. The polar diameter (PD) and equatorial diameter (ED) of each fruit were measured with a digital caliper.

### 2.7. Mineral Content of the Fruit

The content of macromineral elements was carried out with a nitric acid digestion technique: 0.5 g of dry sample was weighed and placed in a 100 mL beaker, then 30 mL of concentrated $HNO_3$ were added and boiled for 3 h while maintaining the initial volume of 30 mL. The digested samples were filtered with a No. 42 Whatman® filter paper (Merck KGaA, Darmstadt, Alemania) and made up to 50 mL with deionized water. The resulting solution was analyzed with a PerkinElmer® spectrometer (Proinstra S.A., Quito, Ecuador), model PinAAcle 900H, at the UAT. The quantification of minerals was carried out for N, K, Ca, Mg, Fe, Cu, Zn, and Mn. The quantification of phosphorus was carried out with the technique of "Phosphorus in plant by UV-Visible spectrophotometry" by using 2 mL of heptamolybdate–vanadate and the amount of deionized water required to make 10 mL. Shake and let it rest for half an hour. Read absorbance of each of the solutions at 470 nm in a UV–visible spectrophotometer.

### 2.8. Statistic Analysis

For data processing, an analysis of variance and a comparison of means were performed using the LSD test ($p \leq 0.05$). Infostat® version 2017 software was used.

## 3. Results

### 3.1. Crop Parameters for Grafts

With the grafting factor, a yield of up to 5.2 kg per plant was achieved, which means an increase of 30% compared to the control, while an increase in the agronomic variables evaluated was observed in PH, NL, NF, PD, ED, and FW, which showed significant differences at early stages compared to the non-graft factor. The implementation of grafted plants showed an increase in fruit weight of 19%, number of fruits by 18%, and fruit firmness by 4%, considering that the modalities of Graft/No Graft did not show statistically significant differences in the concentration of pH (potential hydrogen) and TSS (total soluble solids).

### 3.2. Crop Parameters for Doses

Fertilized treatments did not show an increase in PH, NL, NF, PD, ED, and FW (Table 1), but in the interaction of the factors, there was an influence of the variable of ED increasing with the treatments 9-0-0, 0-12-0, and 0-0-9, up to 4%, which favors the application of the doses. The 9-0-0 dose participates in the 13% increase with respect to the other doses.

With all the doses, the fruit weight was increased compared to the absolute control, up to 6% for the 0-12-0, 0-0-9, and 9-12-9 doses and 5% for the dose 9-0-0.

**Table 1.** Effect of the application of calcium, potassium, and magnesium under agronomic parameters and commercial quality in tomato fruits.

| | Treatment | PH | NL | NF | PD | ED | FW |
|---|---|---|---|---|---|---|---|
| Factor 1: Graft | Grafted | 179.68 a | 25.04 a | 32.72 a | 87.61 a | 55.51 a | 158.1 a |
| | Ungrafted | 157.60 b | 22.60 b | 29.76 b | 81.61 b | 48.93 b | 134.22 b |
| Factor 2: Fertilizer dose Ca-K-Mg | 0-0-0 | 167.50 a | 23.80 a | 31.50 a | 84.56 a | 51.79 b | 139.54 b |
| | 9-0-0 | 158.50 a | 23.90 a | 31.90 a | 85.15 a | 53.37 a | 146.95 a |
| | 0-12-0 | 160.60 a | 23.70 a | 32.00 a | 84.15 a | 52.08 b | 148.51 a |
| | 0-0-9 | 169.70 a | 23.90 a | 31.40 a | 84.29 a | 51.98 b | 148.30 a |
| | 9-12-9 | 168.9 a | 23.80 a | 31.90 a | 84.66 a | 51.88 b | 147.28 a |
| Interaction | | | | | | | |
| Grafted | 0-0-0 | 178.00 a | 22.60 b | 33.60 a | 87.00 b | 54.72 b | 152.94 b |
| Grafted | 9-0-0 | 179.80 a | 25.00 a | 33.80 a | 89.08 a | 57.98 a | 160.10 a |
| Grafted | 0-12-0 | 179.80 a | 24.80 a | 33.60 a | 86.44 b | 55.28 b | 159.60 a |
| Grafted | 0-0-9 | 180.00 a | 25.40 a | 33.40 a | 87.16 b | 55.20 b | 159.58 a |
| Grafted | 9-12-9 | 180.80 a | 25.00 a | 34.20 a | 87.88 a | 54.38 b | 158.48 a |
| Ungrafted | 0-0-0 | 157.00 b | 22.60 b | 29.40 b | 82.12 c | 48.86 c | 126.74 e |
| Ungrafted | 9-0-0 | 157.20 b | 22.80 b | 30.00 b | 81.22 c | 48.76 c | 133.80 d |
| Ungrafted | 0-12-0 | 157.40 b | 22.60 b | 30.40 b | 81.86 c | 48.88 c | 137.54 c |
| Ungrafted | 0-0-9 | 159.40 b | 22.40 b | 29.40 b | 81.42 c | 48.78 c | 136.92 c |
| Ungrafted | 9-12-9 | 157.00 b | 22.60 b | 29.60 b | 81.44 c | 49.38 c | 136.08 c |
| | CV (%) | 1.88 | 4.18 | 5.32 | 1.76 | 2.6 | 1.5 |

PH (cm) = plant height, NL = number of leaves, NF: number of fruits, PD (mm): polar diameter of fruit, ED (mm): equatorial diameter of fruits, FW (g) = weight of fruits/plant. Means with the same letter are not significantly different ($p \leq 0.05$), ($p > 0.05$, Fisher's protected LSD test), CV (%) = coefficient of variation.

### 3.3. Crop Parameters for Interactions

Fertilizer dose × graft interaction had an increase in all doses where a grafting plant was used with respect to plant height, number of leaves, number of fruits, and fruit weight, highlighting an improvement for the dose where potassium fertilization was not used. The fruits' polar and equatorial diameters were positively influenced by the treatments with a supplementary application of calcium, and all the doses where at least one fertilizer applied is used in grafted plants increase the weight of the fruit by up to 17%, as can be seen in Table 1.

### 3.4. Mineral Profile of the Fruit of Grafted Plants

The implementation of grafted plants highly significantly increased the mineral contents of fresh fruits in Ca with an increase of 10%, P 1%, Mn 6%, Zn 7%, Cu 6%, Fe 64%, K 21%, and Mg 19% (Table 2).

### 3.5. Results of the Mineral Profile of the Fruit for the Different Doses

Table 2 shows that the dose of 9-12-9 milliequivalents statistically increased the percentage of minerals, such as Ca 6%, P 4%, Mn 12%, Zn 14%, Cu 8%, Fe 74%, and Mg 25%.

The interaction with grafted plants and Mg positively influenced the mineral content of the tomato fruits, although if other elements are added to this dose, as in the dose of 9-12-9 milliequivalents of Ca, K, and Mg, it gives as a result an increase in the mineral composition of the fruits of the grafted plants.

**Table 2.** Effect the application of calcium, potassium, and magnesium under mineral profile in tomato fruits.

| | Treatment | N | P | K | Ca | Mg | Fe | Cu | Zn | Mn |
|---|---|---|---|---|---|---|---|---|---|---|
| | | % | mg/kg | | | | | | | |
| Factor 1: Graft | Grafted | 2.43 a | 352.00 a | 3359.00 a | 166.07 a | 123.60 a | 49.15 a | 1.97 a | 6.92 a | 4.50 a |
| | Ungrafted | 2.32 b | 348.00 b | 2937.00 b | 151.20 b | 104.30 b | 29.85 b | 1.85 b | 6.45 b | 4.24 b |
| Significance | | 0.0187 | 0.0001 | 0.00001 | 0.0001 | 0.0001 | 0.0001 | 0.0001 | 0.0001 | 0.0001 |
| Factor 2: Fertilizer dose Ca-K-Mg | 0-0-0 | 2.23 c | 345.00 c | 3092.00 a | 155.37 b | 102.80 b | 34.10 c | 1.84 c | 6.41 b | 4.20 c |
| | 9-0-0 | 2.34 b | 345.00 c | 3216.00 a | 155.55 b | 105.10 b | 34.30 c | 1.90 b | 6.64 b | 4.38 b |
| | 0-12-0 | 2.48 a | 345.00 c | 3113.00 a | 153.00 b | 103.80 b | 34.17 c | 1.86 b | 6.52 b | 4.29 b |
| | 0-0-9 | 2.30 c | 352.00 b | 3120.00 a | 164.17 a | 129.10 a | 35.19 b | 1.93 a | 6.42 b | 4.27 b |
| | 9-12-9 | 2.52 a | 359.00 a | 3198.00 a | 164.29 a | 129.00 a | 59.74 a | 1.99 a | 7.46 a | 4.73 a |
| Significance | | 0.0019 | 0.0001 | 0.3615 | 0.0001 | 0.0001 | 0.0001 | 0.041 | 0.0001 | 0.0001 |
| Interaction | | | | | | | | | | |
| Grafted | 0-0-0 | 2.19 c | 348.00 b | 3306.00 a | 158.61 b | 103.60 b | 35.45 c | 1.86 c | 6.56 b | 4.21 c |
| Grafted | 9-0-0 | 2.44 b | 347.00 b | 3446.00 a | 159.26 b | 104.50 b | 35.64 c | 1.93 b | 6.76 b | 4.41 b |
| Grafted | 0-12-0 | 2.55 a | 348.00 b | 3330.00 a | 156.26 b | 103.60 b | 36.46 c | 1.89 c | 6.44 b | 4.25 b |
| Grafted | 0-0-9 | 2.44 b | 358.00 a | 3311.00 a | 178.09 a | 154.00 a | 40.66 b | 2.02 a | 6.51 b | 4.34 b |
| Grafted | 9-12-9 | 2.55 a | 360.00 a | 3402.00 a | 178.15 a | 152.50 a | 89.52 a | 2.13 a | 8.34 a | 5.29 a |
| Ungrafted | 0-0-0 | 2.27 b | 343.00 c | 2878.00 b | 152.14 c | 102.20 b | 29.74 d | 1.83 c | 6.25 c | 4.18 c |
| Ungrafted | 9-0-0 | 2.24 b | 344.00 b | 2987.00 b | 151.84 c | 105.70 b | 29.96 d | 1.88 c | 6.52 b | 4.34 b |
| Ungrafted | 0-12-0 | 2.40 b | 347.00 b | 2897.00 b | 151.35 c | 104.00 b | 29.85 d | 1.83 c | 6.44 b | 4.32 b |
| Ungrafted | 0-0-9 | 2.17 c | 346.00 b | 2929.00 b | 150.25 c | 104.20 b | 29.72 d | 1.84 c | 6.33 c | 4.21 c |
| Ungrafted | 9-12-9 | 2.49 a | 358.00 a | 2995.00 b | 150.42 c | 105.50 b | 29.96 d | 1.85 c | 6.57 b | 4.16 c |
| Significance | | 0.1547 | 0.0013 | 0.9886 | 0.0001 | 0.0001 | 0.0001 | 0.0129 | 0.0001 | 0.0001 |
| | CV (%) | 6.17 | 0.8 | 4.7 | 1.56 | 4.1 | 0.66 | 4.13 | 3.78 | 3.1 |

N: nitrogen, P: phosphorus, K: potassium, Ca: calcium, Mg: magnesium, Fe: iron, Cu: copper, Zn: zinc, and Mn: manganese. Means with the same letter per column and within each statistical partition: graft, dose, and interaction are not statistically different ($p \leq 0.05$), LSD Fisher test. CV (%) = coefficient of variation.

## 4. Discussion

Grafting is a widely used technology in modern agriculture, and it plays an important role in the propagation of vegetative growth, renewing varieties, improving the absorption of nutrients, regulating the accumulation of ions, and improving the resistance of plants [16]. Regarding the factors evaluated, the results obtained in this study (Table 1) showed an increase in each of the variables that differentiate the two factors of grafted tomato plants versus non-grafted tomato plants, as well as in plant height, number of leaves, number of fruits, fruits' weight, and equatorial and polar diameters of the fruit, because a grafted plant is more efficient in the absorption and transport of water and ions from the root to the other organs, increasing the accumulation of biomass [16]. Treviño López [17] found that the effect of grafting has a positive effect on production variables, such as an increase in fruit diameter and weight compared to plants without grafting. Similar results to those of a research work where they reported an increase in fruit length, number of fruits, and yield per plant, the highest values were obtained in fruits of grafted plants [4]. The vegetative growth of grafted plants is largely manifested by the number of leaves. In a study carried out with cucumbers, differences between treatments were observed in favor of the grafted, and the trend was that its value increased by 27% with the graft [18]. The results regarding the number of leaves concur with those obtained by García-Ávila, where it was reported a greater specific leaf area (SLA) in grafted pumpkin plants than in non-grafted ones [10]. Plant height is an important variable, although this effect varies according to the cultivar studied [16]. The plant height variable in this research increased up to 14% compared to non-grafted plants; therefore, statistically significant differences were observed. A recent work evaluated plant height in grafted plants, and an increase was observed compared to non-grafted plants, where this variable is reflected as a plant with greater vigor that may be feasible for the translocation of elements through the plant's pathways [5]. The size of the fruits of the grafted plants increased, and this can occur depending on the rootstock

used [19]. Similar results were presented by Grimaldo Juárez [20], where the average fruit size was greater in grafted plants, compared to the normal condition, with an increase of 17.7%, respectively. A study carried out on bell pepper shows that the Terrano rootstock increased yield by 50% [7]. The accumulation of K in grafted plants is attributed to physical characteristics of the root system, such as abundant lateral and vertical root development, exploration area, and, consequently, greater absorption of water and minerals [21]. A study carried out showed that there is an increase in the mineral content of grafted plants because of a higher concentration of the elements and accumulation of biomass [8]. Regarding the fruit weight variable, it is necessary to highlight that all the applications of the treatments where at least one dose of fertilizers was supplied increased fruits' weight compared to the control. As shown in Table 2, regarding the treatments, the best ones were where the interaction of each of the doses with the grafted plants was obtained, similar to that reported by Pérez-Grajales [22] where different treatments were tested, and all the treatments where the graft with the tomato variety CID was used improved the agronomic variables, increased yield, total biomass accumulation, and mineral composition. The two factors used show that the grafts have statistically significant differences in the mineral composition of tomato fruits due to the nutrient assimilation capacity provided by this technique [6]. When there is an adequate supply of potassium, the fruit tends to store more of one mineral than another due to the competition that exists between minerals. In conditions of potassium excess in the soil, the plant increases its consumption, making it able to also modify the absorption and availability of other minerals, such as calcium and magnesium, which can favor the fruits [14]. For the potassium content in the fruits, there were no statistically significant differences. The grafts increased the values of the agronomic variables, with higher values in variables such as leaf area. It may be said that the leaf area and height of the plant, which resulted in an increase of 0.96% in this variable, is reflected in a yield of 38% compared to the control [17]. A study shows that tomato fruits from grafted plants with Multifort/CID modified the mineral composition and improved yield [8]. Although the best dose was the supplementary addition of 9-12-9, with an increase in Ca, P, Mn, Zn, Cu, Fe, K, and Mg due to the interaction Ca-K-Mg (adsorption–desorption), these nutrients are considered poorly mobile. K is required in large quantities by crops (similar to N), while Ca and Mg are required in smaller quantities (similar to P and S) [10], which were made available for the fruits in this research. Therefore, the best interaction regarding the agronomic variables was obtained using the 9-0-0 dose where only Ca was applied in the grafted plants, similar to that reported by Rodríguez-Mendoza [23], who obtained favorable results in calcium applications in agronomic variables.

## 5. Conclusions

The grafting factor directly affects the agronomic variables of the tomato plant, where it mainly increases the diameter and weight of the fruit, which increases the total yield. Fruit quality variables and fruit mineral profile were also positively affected by the grafting factor. Ca-K-Mg applications affect production variables by increasing the number of leaves, number of fruits, polar diameter of the fruits, and weight. In fruit quality, grafting with the recommended dose (9-12-9) increases fruit firmness and fruit TSS; it also participates in the two doses where calcium is used: 9-12-9 and 9-0-0. In the mineral profile of the fruits, the grafted plants participate directly in the increase and stability of the fruits. Production increased with the interaction of factors in fruit weight, number of fruits, and polar and equatorial diameters. Due to the results obtained in this work, we recommend the use of grafts and the foliar application of Ca-K-Mg, which could be used in tomato production to increase productivity and the mineral profile of the fruit.

**Author Contributions:** Conceptualization, M.C.D.l.F. and R.M.P.M.; methodology, R.D.M.; software, A.J.M.; validation, A.J.M. and R.M.P.M.; formal analysis, M.C.D.l.F.; investigation, A.B.M.; resources, R.D.M.; data curation, M.C.D.l.F.; writing—original draft preparation, R.M.P.M.; writing—review and editing, M.C.D.l.F.; visualization, A.B.M.; supervision, M.C.D.l.F.; project administration,

M.C.D.l.F.; funding acquisition, M.C.D.l.F. All authors have read and agreed to the published version of the manuscript.

**Funding:** This research was funded by University Autonomus Agrarian Antonio Narro, grant number 3146.

**Institutional Review Board Statement:** Not applicable.

**Data Availability Statement:** The data presented in this study are available in article.

**Acknowledgments:** To the supporting research staff at Universidad Autónoma Agraria Antonio Narro and the post-doctorate scholarship by Mexico from CONAHCyT.

**Conflicts of Interest:** The authors declare no conflict of interest.

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
