# Peer review of "Calcium, Potassium, and Magnesium Affect the Nutritional Value of Tomato Grafted Fruits Grown in a Nutrient Film Technique System"

_agriculture, doi:10.3390/agriculture13122189_

Round 1
Reviewer 1 Report
Comments and Suggestions for Authors
Thank you for considering me in evaluating this manuscript. This is a study involving two tomato production factors in hydroponics-NFT (grafting and Ca-K-Mg concentration). I consider the justification for the study weak, although there are two very important factors in the production of this vegetable. I made several comments on the manuscript.

Author Response
The observations was attended, you can see in the attached file

Reviewer 2 Report
Comments and Suggestions for Authors
Dear authors,
The topic of this paper is worth researching, but there are many questions in the writing, so this paper needs major modification.
The suggestions can be seen from the side comments in the manuscript reviewed back, and suggest to modify the manuscript carefully.

Author Response
The observations was attedned, in the file attached you can see.

Reviewer 3 Report
Comments and Suggestions for Authors
Dear Authors,
The title of the article should be changed.
Fill in keywords (for example tomato) and write in alphabetical order.
Why the references cited in the article are not from the first source, and so on.
Justify the necessity of the study. Why this study is relevant and necessary?
Row 95 - write (Figure 1) instead of (figure 1).
You can clarify the experimental scheme: a0- grafted, a1 – ungrafted, b0 - 0, b1 – Ca-K-Mg 9-0-0, etc.
120, 122 and 127 rows do not write abbreviations (NL, etc.).
In row 151 you write „a yield of up to 5,2 kg per plant“, but in Table 1 you write g: „FW(g) weight of fruits/plant“.
Rows 179-183 and 202-206 need to be removed.
Row 189 not „Table 2 shows“, but „according to research data“.................
191 row - sentence not required?
Change table titles („Effect of tomato grafting and Ca-K-Mg dosage on.....“)
This: „PH (cm) = plant height, NL = number of leaves, NF: num-175 ber of fruits, PD (mm): polar diameter of fruit, ED (mm): equatorial diameter of fruits, FW (g) = 176 weight of fruits/plant. Means with the same letter are not significantly different (p ≤ 0.05), LSD Fisher 177 test), C. V = coefficient of variation.“ to move under the table.
Write PH - plant height, cm, but no PH (cm) = plant height, etc.
This:“ Means with the same letter per column and within each statistical partition: graft, dose and interaction are not statistically different (p ≤ 0.05), LSD Fisher test. C. V. = Coefficient of Variation, N: Nitrogen, P: Phosphorus, K: Potassium, Ca: Calcium, Mg: Magnesium, Fe: Iron, Cu: Copper, Zn: Zinc, and Mn: manganese.“ to move under the table.
What absolute control is, explain.
Row 281 is repeated, „in this“.
In rows 229-231 do not write abbreviations (PH, etc.).
The chapters of the article shall be arranged in order.

Author Response
The observations was attended, you can see in the attached file.
